Age, but not short-term intensive swimming, affects chondrocyte turnover in zebrafish vertebral cartilage

Jian Quan-Liang 1
HuangFu Wei-Chun 2
Lee Yen-Hua 1
http://orcid.org/0000-0002-4524-3263 Liu I-Hsuan 1 3 4 ihliu@ntu.edu.tw
1 Department of Animal Science and Technology, National Taiwan University , Taipei , Taiwan
2 The Ph.D. Program for Cancer Biology and Drug Discovery, College of Medical Science and Technology, Taipei Medical University , Taipei , Taiwan
3 Research Center for Developmental Biology and Regenerative Medicine, National Taiwan University , Taipei , Taiwan
4 School of Veterinary Medicine, National Taiwan University , Taipei , Taiwan
Bienzle Dorothee
Electronic publication date: 2018 Oct 1
Publication date: 2018
Volume: 6
Electronic Location ID: e5739
Received 2018 Jan 22; Accepted 2018 Aug 30
Copyright: © 2018 Jian et al.
Copyright year: 2018
Copyright holder: Jian et al.
License: This is an open access article distributed under the terms of the Creative Commons Attribution License, which permits unrestricted use, distribution, reproduction and adaptation in any medium and for any purpose provided that it is properly attributed. For attribution, the original author(s), title, publication source (PeerJ) and either DOI or URL of the article must be cited.
License URL: https://creativecommons.org/licenses/by/4.0/

Keywords: Aging, Cartilage, Osteoarthritis, Tissue homeostasis, Chondrocyte

Funding: The Ministry of Science and Technology of Taiwan 105-2628-B-002-005-MY4 Council of Agriculture of Taiwan 107AS-22.1.6-AD-U1(13) National Taiwan University 106R7724 Taipei Medical University 105TMU-CIT-01-1 This study was financially supported by the Ministry of Science and Technology of Taiwan (105-2628-B-002-005-MY4, to I-Hsuan Liu), Council of Agriculture of Taiwan (107AS-22.1.6-AD-U1(13), to I-Hsuan Liu), National Taiwan University (106R7724, to I-Hsuan Liu) and Taipei Medical University (105TMU-CIT-01-1, to Wei-Chun HuangFu). The funders had no role in study design, data collection and analysis, decision to publish, or preparation of the manuscript.

==============================
Both age and intensive exercise are generally considered critical risk factors for osteoarthritis. In this work, we intend to establish zebrafish models to assess the role of these two factors on cartilage homeostasis. We designed a swimming device for zebrafish intensive exercise. The body measurements, bone mineral density (BMD) and the histology of spinal cartilages of 4- and 12-month-old zebrafish, as well the 12-month-old zebrafish before and after a 2-week exercise were compared. Our results indicate that both age and exercise affect the body length and body weight, and the micro-computed tomography reveals that both age and exercise affect the spinal BMD. However, quantitative analysis of immunohistochemistry and histochemistry indicate that short-term intensive exercise does not affect the extracellular matrix (ECM) of spinal cartilage. On the other hand, the cartilage ECM significantly grew from 4 to 12 months of age with an increase in total chondrocytes. dUTP nick end labeling staining shows that the percentages of apoptotic cells significantly increase as the zebrafish grows, whereas the BrdU labeling shows that proliferative cells dramatically decrease from 4 to 12 months of age. A 30-day chase of BrdU labeling shows some retention of labeling in cells in 4-month-old spinal cartilage but not in cartilage from 12-month-old zebrafish. Taken together, our results suggest that zebrafish chondrocytes are actively turned over, and indicate that aging is a critical factor that alters cartilage homeostasis. Zebrafish vertebral cartilage may serve as a good model to study the maturation and homeostasis of articular cartilage.

Introduction

Osteoarthritis (OA) is the most common pathologic condition of articular cartilage and leads to joint pain and stiffness, degeneration of articular cartilage and, sometimes, ectopic osteogenesis to form osteophytes. It is generally believed that both age and mechanical loading are critical risk factors for OA. Accordingly, previous reports indicate that age is positively correlated to the prevalence of OA and elite athletes have a higher risk of OA and arthroplasty (Busija et al., 2010; Tveit et al., 2012). More importantly, articular cartilage poorly regenerates, and OA is generally considered only treatable, but incurable.

Histologically, cartilage can be categorized into three major types: hyaline cartilage, fibrocartilage and elastic cartilage. Articular cartilage is hyaline cartilage, which predominantly contains a homogenous and translucent extracellular matrix (ECM) and is covered by perichondrium. The ECM in the articular cartilage is mostly type II collagen with some other collagens, proteoglycans and glycosaminoglycans (GAGs) such as hyaluronan, chondroitin sulfate and keratan sulfate. The cells only account for a small portion of the volume in articular cartilage. In mammalian joints, the articular cartilage can be divided into four regions including a superficial zone, middle zone, deep zone and calcified zone. The superficial zone contains the highest cell density and the superficial cells secret proteoglycan 4 as a joint lubricant. These cells have a large long/short morphological axis ratio (Schumacher et al., 1994). The middle zone accounts for the major volume of an articular cartilage and the middle cells are enlarged, with an oval shape, usually sitting in lacunae compared to the superficial cells (Hedlund et al., 1999). The deep cells are usually round hypertrophic chondrocytes, while the calcified zone is a transition region between cartilage and subchondral bone (Grogan et al., 2009; Schmid & Linsenmayer, 1985).

Due to its avascular and aneural nature, articular cartilage was historically believed to be inert. Later, injection of radioactive isotopes indicates a dynamic change in the composition of GAGs and indicates that cartilage is not completely lack of metabolism (Davidson & Small, 1963; Mankin & Lippiello, 1969). Studies taking advantage of the fluctuation of atmospheric 14C support the notion that GAGs are dynamically turned over, but also reveal that collagen in the articular cartilage is extremely inert (Heinemeier et al., 2016; Libby et al., 1964). Moreover, studies in the recent decade suggest that superficial cells might work as stem cells that supply new chondroblasts for cellular turnover in the articular cartilage (Alsalameh et al., 2004; Candela et al., 2014; Dowthwaite et al., 2004). It is now widely believed that cellular turnover supported by endogenous stem cells occurs in the articular cartilage during young ages but not in mature articular cartilage, which might result in the age-related risk for OA.

Since buffering mechanical loading is one of the physiological functions of articular cartilage, it is reasonable to expect articular cartilage to bear a certain mechanical load. Joint cartilages that are immobilized for weeks to months result in signs of OA including loss of GAGs and formation of osteophytes (Jurvelin et al., 1985; Langenskiold, Michelsson & Videman, 1979; Videman, Eronen & Friman, 1981). On the other hand, not only professional athletes have a higher prevalence of OA, but articular cartilage in animals that receive rigorous exercise training also show OA-like changes (Arokoski et al., 1993; Kujala, Kaprio & Sarna, 1994; Lequesne, Dang & Lane, 1997; Paukkonen et al., 1985; Saamamen et al., 1994; Tveit et al., 2012). Although the mechanisms for mechanical loading to affect articular cartilage remains elusive, multiple lines of evidence suggest that a moderate level of mechanical loading is beneficial to the articular cartilage (Kiviranta et al., 1988; Saamanen et al., 1990).

Zebrafish have emerged as an excellent model to study embryonic development as well as tissue regeneration, but they could also serve as a model to study homeostasis and aging. Previous study indicates that ability and trainability of physical exercise declines with the age of zebrafish, similar to mammals (Gilbert, Zerulla & Tierney, 2014). Furthermore, not only is vertebral cartilage development and maturation promoted by exercise training, aging also leads to deformity of vertebral cartilage that recapitulates signs of OA (Fiaz et al., 2012; Hayes et al., 2013). In this study, we aimed to determine whether age and exercise training affect the homeostasis of vertebral cartilage in adult zebrafish by evaluating the content of GAGs and type II collagen as well as cellular dynamics.

Materials and Methods

Zebrafish strain and maintenance

The AB wild-type line of zebrafish purchased for exercise experiments (GenDanio Aquaculture System, New Taipei City, Taiwan) were randomly segregated into two groups: control (Ctrl) vs. exercise (Exe), and all the comparisons between before (+0 days) vs. after (+14 days) exercise training program were done with these zebrafish. The AB wild-type line of zebrafish at 3–4 months of age and 11–13 months of age were purchased (Azoo Co., Taipei, Taiwan) for the age comparisons (4- vs. 12-month-old). All zebrafish were kept individually (in 200 mL water) at 28.5 °C with a light cycle of 14-h light/10-h dark and were fed twice daily for this study (Chang et al., 2016; Westerfield, 2000). All experimental procedures in this study were reviewed and approved by the Institutional Animal Care and Use Committee of National Taiwan University (NTU105-EL-00037) and were performed in accordance with the approved guidelines.

Zebrafish intensive exercise training

To force zebrafish to go through intensive exercise training, a simple training system was designed and assembled with an aquatic powerhead (Rio+1400; Technological Aquatic Association Manufacturing, Thousand Oaks, CA, USA) connected to a polyisoprene tube (52 cm in length, 2.5 cm in diameter) and a mesh covering the end opening (Fig. 1A). Each system housed an individual zebrafish in the tube during the training session and a 60-L polyethylene tank housed three training systems placed in a stack fashion.

Figure 1 Both age and intensive exercise affect growth of adult zebrafish.

(A) An aquatic powerhead was connected to a clear water pipe to enforce intensive exercise in adult zebrafish. (B) The schedule for zebrafish underwent exercise-training consist of an 8-h exercise training session (red) two feeding sessions (blue) during the 14-h light period and a resting period during the 10-h dark period (black). (C) The body length of 12-month-old zebrafish (n = 12) grew significantly compared to the 4-month-old zebrafish (n = 8; Mann–Whitney’s test). Data are presented as mean ± SEM. (D) After intensive exercise (Exercise) for 14 days, body length of 12-month-old zebrafish was significantly shorter, while the zebrafish in control group (control) was not (Wilcoxon matched-pairs signed rank test). (E) The body weight of 12-month-old zebrafish (n = 12) grew significantly compared to the 4-month-old zebrafish (n = 8; Mann–Whitney’s test). Data are presented as mean ± SEM. (F) The body weight continued to grow in 14 days in 12-month-old (Control), but intensive exercise (Exercise) hindered this growth (Wilcoxon matched-pairs signed rank test). n.s., not significant (P < 0.05); *P < 0.05; **P < 0.01; ***P < 0.001; ****P < 0.0001.

Each zebrafish in the exercise group was assessed for their maximal resisting speed. Briefly, the flow of the aqua pump was increased every minute until the zebrafish fail to resist and reside the mesh (fail speed). The maximal speed (i.e., the flow speed immediately before the fail speed) was then calculated according to the milliliter/minute of the aqua pump and the cross-sectional area of the tube (Table 1). Each zebrafish in the exercise group was transferred into the system 30 min after the morning feed and rested for 30 min before the 8-h training session at maximal speed began (Fig. 1B). After the training session, the zebrafish was transferred back to the housing system, received an excessive night feed. The training session lasted for 14 days, and the control group in this experiment was managed in the same way without turning the aqua pump on.

Table 1 The maximal speeds of exercising zebrafish.

Zebrafish	Flow speed (cm/s)	
no. 4	23.9	
no. 5	23.2	
no. 8	24.0	
no. 10	25.1	
no. 13	23.4	
no. 15	22.2	
no. 16	22.0	
no. 17	23.7	

Body measurements

To assess the effects of age and exercise training on overall physiological condition, body weight and body length of each zebrafish were measured and recorded. Briefly, each zebrafish was anesthetized in 0.016% ethyl 3-aminobenzoate methanesulfonate (MS-222; Sigma-Aldrich, St. Louis, MO, USA) before the morning feed. The body weight was measured on a precision balance (PJ3600; Mettler-Toledo, Columbus, OH, USA) after excessive water was removed and a photograph was taken to measure the body length from the mouth tip to the end of the tail-fin using ImageJ (Schneider, Rasband & Eliceiri, 2012).

Micro-computed tomography and bone mineral density

To assess the effects of age and exercise training on the skeletal system, bone mineral density (BMD) of each zebrafish was estimated using images from micro-computed tomography (microCT). Briefly, zebrafish were anesthetized using a mixture of 100 ppm MS-222 and 100 ppm isoflurane (Abbott Laboratories, Queenborough, UK) in water (Huang et al., 2010), restrained between two wet sponges, and scanned (SkyScan-1076; Bruker microCT, Kontich, Belgium) with nine μm resolution, 80 kV, 124 μA, 0.5° rotational step, 1,700 ms exposure and a 0.5 mm aluminium filter (Table S1). To standardize and calibrate the intensity, two scanning phantoms with 0.25 and 0.75 g/cm3 of densities were used. The scale was designed according to a pre-defined parameter (air, HU = −1,000, color index = 0) and scanning results (including water and phantoms) to generate the mapping reference between color index (0–255) and Hounsfield units (HUs; in our case, −1,000 to 3,184). The 3D rendering was done by using CTvox software (Bruker microCT, Kontich, Belgium) and the BMD was analyzed by using CTAn software (Bruker microCT, Kontich, Belgium) with the fourth (a Weberian vertebra) or all vertebrae as the region of interest (Bird & Mabee, 2003; Hur et al., 2017).

Labeling and tracing of 5-bromo-2’-deoxyuridine (BrdU)

To understand the effect of age and short-term intensive exercise training on chondrocyte proliferation, 5-bromo-2’-deoxyuridine (BrdU; Sigma-Aldrich, St. Louis, MO, USA) was used to label the proliferative cells. For age comparisons, the zebrafish were anesthetized in 0.016% MS-222 3 h after the night feed with five μL of BrdU (2.5 mg/mL in distilled water) were administered via oral gavage once a day for consecutive 15 days (Reimer et al., 2008). To reduce stress for the exercise comparisons, zebrafish were immersed in 200 mL of BrdU (150 μg/mL) each day during the dark period of the 2-week training session (Rowlerson et al., 1997). To determine whether quiescent cells exist in the cartilage, the BrdU labeled zebrafish were chased for an additional 30 days.

Histology preparation

To observe the effect of age and short-term intensive exercise training on the morphology and composition of cartilage, qualitative and quantitative histology were analyzed. Briefly, the zebrafish were sacrificed in 20 mL 0.4% MS-222, and fixed in 20 mL 4% paraformaldehyde (Merck, Darmstadt, Germany) at 4 °C for 5 days after the abdomen was opened with a scalpel for better penetration of the fixative. The zebrafish were then immersed in 20 mL of 10% ethylenediaminetetraacetic acid (EDTA, Amresco, Solon, OH, USA) for 3 days for de-calcification. After the fixatives and EDTA were washed away with water, dehydration was with 20 mL each of a 30–100% ethanol gradient. The sample was then immersed two times in 20 mL of xylene (J.T. Baker, Center Valley, PA, USA) for 1 h each and embedded in paraffin (Surgipath Medical Industries, Richmond, IL, USA). The tissue blocks were sectioned at five μm to produce consecutive sagittal sections using a rotary microtome (HM315; Microm, Walldorf, Germany). The tissue slides were then rehydrated using xylene followed by a 100–30% ethanol gradient, and finally immersed two times in phosphate buffered saline (PBS; Amresco, Solon, OH, USA) for 5 min. From the most lateral edge of the vertebral column to the midline, about 30 (in 4-month-old) or 40 (in 12-month-old) consecutive sections could be obtained. For all quantitative image analysis, five sections from the same subject with consistent 20 μm (in 4-month-old) or 25 μm (in 12-month-old) interval between sections were used for the same staining analysis and the sum of the results from five slides represented the result for the subject.

Histochemistry

To observe the GAG content in the cartilage, the tissue slides were immersed in hematoxylin (Surgipath Medical Industries, Richmond, IL, USA) for 5 min, washed with 1% HCl (Merck, Darmstadt, Germany) and distilled water, immersed in 0.02% fast green (Merck, Darmstadt, Germany) for 1 min, washed with 1% acetate (Merck, Darmstadt, Germany) and distilled water, immersed in safranin O (Merck, Darmstadt, Germany) for 10 min, washed with 95% and 100% ethanol, and finally sealed with mounting medium (Muto Pure Chemicals, Tokyo, Japan).

Immunohistochemistry

To observe the distribution of type II collagen and BrdU labeling/retention in the cartilage, immunohistochemistry was performed. Briefly, after the tissue slides were rehydrated, epitope retrieval was achieved by proteinase K (20 μg/mL; GeneMark, Taichung, Taiwan) incubation at room temperature for 30 min (for type II collagen). For BrdU, incubation was with sodium citrate 2.94 mg/mL, pH 6.5; Sigma-Aldrich, St. Louis, MO, USA) at step-up temperatures from 65 to 95 °C for 10 min. After washing with 0.1% Phosphate Buffered Saline (PBS) with Tween-20 (PBST) (Amresco, Solon, OH, USA) twice, the tissue slides were blocked with blocking buffer (3% bovine serum albumin (Sigma-Aldrich, St. Louis, MO, USA) in PBS) at room temperature for 30 min and incubated with the primary antibodies against type II collagen (1:10 in blocking buffer; Developmental Studies Hybridoma Bank, Iowa City, IA, USA) for 2.5 h or against BrdU (1:100 in blocking buffer; AbD Serotec, Kidlington, UK) for 30 min at room temperature. After washing with 0.1% Tween-20 in PBS, the primary antibodies were detected by goat anti-mouse IgG conjugated with Alex Fluor 555 (1:300 in blocking buffer; Abcam, Cambridge, UK) for 1 h at room temperature. The nuclear counter-staining was done by 4′,6-diamidino-2-phenylindole (DAPI, 10 mg/mL; Biotium, Fremont, CA, USA) and the tissue slides were sealed with Fluoroshield mounting medium (Abcam). A primary-free, secondary-only antibody staining was used as a negative control.

Terminal deoxynucleotidyl transferase dUTP nick end labeling (TUNEL)

To assess the effect of age and exercise training on chondrocyte apoptosis, the terminal deoxynucleotidyl transferase (TdT) dUTP nick end labeling (TUNEL) assay was performed using a commercial kit (In Situ Cell Death Detection Kit, TMR red; Roche, Basel, Switzerland). Briefly, after rehydration, the tissue slides were perforated with proteinase K (20 μg/mL) for 15 min at room temperature and washed with PBST. The tissue slides were then incubated with the TdT and dUTP mixture (1:10) at 37 °C for 1 h, counter stained with DAPI, and sealed. A DNase I (two U/μL; Geneaid Biotech, Taipei, Taiwan) treatment at room temperature for 10 min after perforation was used as a positive control, while a TdT-free reaction was used as a negative control for this staining.

Micrographs and image analysis

All the tissue slides were documented by a tissue scanner (TissueFAXS;, TissueGnostics, Vienna, Austria) or by a confocal microscope equipped with differential interference contrast (TCS SP5 II; Leica Microsystems, Wetzlar, Germany). For the quantitative image analysis, three (for type II collagen) or five slides from the same subject with a consistent interval between sections were used for the same staining analysis. The sum of the results represented the subject. The software HistoQuest (TissueGnostics, Vienna, Austria) was used to quantitatively analyze the contents of GAGs and type II collagen in the cartilage, while ImageJ was used to count the nuclei with DAPI, BrdU or TUNEL staining (Schneider, Rasband & Eliceiri, 2012).

Scanning electron microscope

For the scanning electron microscope (SEM) imaging, the tissue sections were collected onto the cover slips (32 × 24 mm). The sections were then de-waxed twice in xylene for 10 min, then three times in 100% ethanol for 10 min each followed by two immersions in acetone for 10 min. Each slide was then critical point dried in liquid CO2 in a critical point dryer (Hitachi, Tokyo, Japan) and ion coated (IB-2, Eiko, Tokyo, Japan) before documentation in the SEM (Inspect S, FEI, Hillsboro, OR, USA).

Statistical analysis

To minimize the potential bias brought about by small sample sizes, non-parametric statistical approaches were used in this study. To minimize the influence due to individual variations, the Wilcoxon matched-pairs signed rank test was used for the comparisons between before and after exercise training sessions. For the comparisons between age groups (4-month-old vs. 12-month-old) and exercise groups (exercise vs. control), the Mann–Whitney U-test was used. For the comparisons among 0-, 15- and 30-day chase, Kruskal–Wallis test was used. Statistical significance was considered when P ≤ 0.05.

Results

Zebrafish continues to grow after sexual maturity while intensive exercise hinders this growth

In mammals, hormones such as estrogen fluctuate dramatically during sexual maturity and trigger the halt in skeletal growth including the closure of epiphyseal plates in bones (Zhong et al., 2011). The body measurements of zebrafish indicate that zebrafish continues to grow after sexual maturity as the body length increased significantly from 2.65 cm at 4 months of age to 3.12 cm (Fig. 1C; Table S2). Intriguingly, although the body length of the control zebrafish was not significantly changed in 2 weeks, the body length of zebrafish was significantly shortened after the 2-week intensive exercise-training program (Fig. 1D; Fig. S1 and Table S2).

Similarly, the body weight was significantly increased from 0.16 g in 4-month-old zebrafish to 0.23 g at 12 months of age (Fig. 1E; Table S3). The body weight continued to increase during the 2-week experimental period in the 12-month-old control group, but the body weight was not altered in the zebrafish experiencing intensive exercise training (Fig. 1F; Table S3). These results indicate that the zebrafish body continues to grow between 4 and 12 months of age, especially the body weight, whereas short-term intensive exercise halts the growth.

The BMD continues to increase after sexual maturity while intensive exercise negatively affects this trend

Previous studies in the human skeletal system indicate that the BMD peaks between 30 and 40 years of age. Appropriate nutrition and exercise enhance the BMD level, but aging especially in females after 50 years of age increases the risk of osteoporosis and OA (Lee et al., 2013; Warming, Hassager & Christiansen, 2002). To assess the potential impact of short-term intensive exercise on the vertebrae column, microCT scan with the fourth vertebrae selected as region of interested was used to estimate the BMD (Figs. 2A and 2B). The result shows that the BMD in the fourth vertebrae continued to grow significantly in the control group during the 2-week experimental period (Fig. 2C; Table S4). The BMD level remained at comparable levels before and after the 2-week intensive exercise training in the exercise group (Fig. 2D; Table S4). These results indicate that zebrafish BMD level in the fourth vertebrae continues to increase even at 12 months of age, but the short-term intensive exercise hinders this increase in BMD.

Figure 2 The accumulation of bone mineral density was affected by intensive exercise.

(A and B) Zebrafish were anesthetized for microCT to evaluate the BMD. A representative transection microCT image of a zebrafish is shown (A). Note that only a cylindrical region (arrows in (A)) containing the hourglass-shaped fourth vertebrae was selected as region-of-interest for quantitative analysis (B). The scale bars represent one mm (A) and 0.15 mm (B). The color scale (0: dark purple; 255: white) represents −1,000 to 3,184 HU. (C and D) The BMD in the fourth vertebrae continued to increase in 12-month-old zebrafish during a 14-day period (C), but intensive exercise hindered this growth trend (D) (Wilcoxon matched-pairs signed rank test). n.s., not significant (P > 0.05); **P < 0.01.

Zebrafish continue to accumulate cartilage ECM after sexual maturity

The GAGs and collagens, especially type II collagen, are the predominant constituents of articular cartilage, and the networking of these macromolecules buffers and disperses the mechanical pressures applied to the joints (Hedlund et al., 1999; Van Der Rest & Mayne, 1988). One of the signatures of OA is the loss of these ECM components (Pritzker et al., 2006). Previous study shows that, as the zebrafish ages, the fourth and fifth vertebrae develop bone and cartilage deformities similar to OA symptoms in humans (Hayes et al., 2013). To assess the effects of age and mechanical loading on ECM accumulation in cartilage, histochemistry, immunohistochemistry and the quantitative analysis of micrographs were performed with the fourth Weberian vertebra (Figs. 3A and 3B) as the region of interest since it includes the largest cartilage ECM, including type II collagen (Fig. 3A) and GAGs (Fig. 3B), compared to other vertebrae. Furthermore, the cartilage at the fourth Weberian vertebra resembles the histological features of hyaline cartilage as it contains no prominent collagen fibers (bracket in Fig. 3C) and is surrounded by fibrous perichondrial organization (arrow and bracket in Fig. 3D).

Figure 3 The fourth Weberian vertebra contained the largest hyaline cartilage among all the zebrafish vertebrae.

Both immunohistochemistry against type II collagen (A) and histochemistry with safranin O and fast green (B) showed that the fourth Weberian vertebra (white arrows in (A)) contained the largest cartilage content in the spine. The scale bar represents 500 μm. (C) The H&E histochemical staining showed typical hyaline cartilage features in the fourth Weberian vertebra. Note a cell-less fibrous region (bracket) is juxtaposited to the cell-rich region of the cartilage. The scale bar represents 50 μm. (D) A representative SEM image showed typical chondrocytes surrounded by lacunae covered by a fibrous perichondrial-like structure (arrow and bracket). The scale bar represents 10 μm. The yellow dotted box in (B) represents the approximate area shown in (C), while the yellow dotted box in (C) represents the approximate area shown in (D).

Interestingly, the distribution of type II collagen was more prominent at the ventral and dorsal end of the cartilage in 4-month-old zebrafish (Fig. 4A), but much more prominent at cartilage margins in 12-month-old zebrafish with no discernible difference between exercise and control groups (Figs. 4B and 4C). Quantitative analysis of immunohistochemical micrographs of type II collagen show significant lower levels in both distribution areas (Fig. 4D; Table S5) and signal density (Fig. 4E; Table S6) in 4-month-old zebrafish compared to 12-month-old zebrafish. On the other hand, both the distribution area (Fig. 4F; Table S5) and signal density (Fig. 4G; Table S6) of type II collagen were at comparable levels in exercise and control groups.

Figure 4 The type II collagen continues to accumulate in the spinal cartilage after sexual maturity.

(A–C) The representative immunohistochemistry fluorescent micrographs showed distinct distributions of type II collagen in the spinal cartilage in 4-month-old (A) and 12-month-old (B and C) zebrafish. The scale bar represents 75 μm. (D–G) Three tissue slides across the sagittal sections of zebrafish vertebrae were obtained with a consistent interval between slides were selected from each subject for quantitative and statistical analysis. Both the occupying area (D) and average density (E) of type II collagen was significantly increased from 4 (n = 7) to 12 months of age (n = 7; Mann–Whitney test). However, the 14-day intensive exercise did not alter the content of type II collagen as both the area (F) and signal density (G) were comparable between the zebrafish in the control group (n = 6) and exercise group (n = 7). Data are presented as mean ± SEM. n.s., not significant (P > 0.05); *P < 0.05; ***P < 0.001.

The histochemistry of safranin O, fast green and hematoxylin can provide well-discerned depictions of GAGs, collagens and cell nuclei in a tissue (Figs. 5A–5C). Compared to type II collagen (Figs. 4A–4C), the GAGs were more prominently distributed in the core of the cartilage, especially in 12-month-old zebrafish, while occupying a larger area in the vertebra (Figs. 5A–5C). Interestingly, although the GAG area size was significantly smaller in 4-month-old zebrafish than in 12-month-old zebrafish (Fig. 5D; Table S7), the signal densities were at a comparable level (Fig. 5E; Table S8). Similar to type II collagen, both the distribution areas (Fig. 5D; Table S7) and signal densities (Fig. 5E; Table S8) of GAGs were at comparable levels between exercise and control groups. Taken together, the short-term intensive exercise training does not result in discernible change in the zebrafish cartilage, while the cartilage continues to grow after the sexual maturity of zebrafish at 4 months of age. Interestingly, the accumulation of type II collagen seems less mature than GAGs in 4-month-old zebrafish, since the signal density of GAGs but not type II collagen remained at comparable levels between 4 and 12 months of age.

Figure 5 The cartilage ECM and chondrocytes varied as the zebrafish grew.

(A–C) The representative histochemical micrographs showed the distribution patterns of GAGs (red as stained by safranin O), collagen (cyan as stained by fast green) and cell nuclei (dark purple as stained by hematoxylin) in the spinal cartilage in 4-month-old (A) and 12-month-old (B and C) zebrafish. The yellow scale bar represents 100 μm. (D–G) Five tissue slides across the sagittal sections of zebrafish vertebrae obtained with a consistent interval between slides were selected from each subject for quantitative and statistical analysis. The occupying area of GAGs (D) was significantly increased from 4 (n = 8) to 12 months of age (n = 8; Mann–Whitney test) without affecting the averaged density (E), but the 2-week intensive exercise-training (n = 8) did not alter the GAG content compared to the control group (n = 8) (Mann–Whitney test). The hematoxylin staining showed that the total chondrocyte number significantly increased from 4 to 12 months of age (G) (Mann–Whitney test) with a significantly decreased cellular density (F) (Mann–Whitney test). However, the 2-week intensive exercise-training did not affect chondrocyte distribution (Mann–Whitney test). Data are presented as mean ± SEM. n.s., not significant (P > 0.05); *P < 0.05; ***P < 0.001.

Cellular dynamics decreased with age

Since the homeostasis of cartilage ECM, especially the GAGs, depend on the balance of catabolism and anabolism of chondrocytes, it is essential to evaluate the chondrocytes in the cartilage. As hematoxylin staining and the existence of lacunae clearly depicted the distribution of chondrocytes and their nuclei (Figs. 5A–5C), the cell counts and cell densities were also evaluated using the same tissue slides. The zebrafish vertebral chondrocytes did not distribute with apparent orientations, but the cells lacking surrounding lacunae were predominately located at marginal regions (Figs. 5A–5C). The 4-month-old zebrafish had significantly greater cell density than the 12-month-old zebrafish (Fig. 5F; Table S9), but the cartilage in 12-month-old zebrafish contained more cells (Fig. 5G; Table S9). Again, both the cell densities and cell counts were at comparable levels between the exercise and control groups. These results indicate that, between 4 and 12 months of age, the continuous growth of cartilage is contributed both by the accumulation of ECM and by the increase in chondrocytes.

To further elucidate whether the increase in chondrocytes was a static accumulation or a result of a dynamic equilibrium, TUNEL staining and BrdU labeling were performed. The TUNEL staining indicates that the apoptotic cells were predominantly located at the outer regions of the vertebral cartilage (most anterior/posterior and dorsal/ventral tips) (Fig. 6A; Table S10). Quantitative analysis shows that cartilage in 12-month-old zebrafish contained a significantly greater percentage of apoptotic cells than cartilage in 4-month-old zebrafish (Fig. 6B), whereas the exercise group was not significantly different from the control group (Fig. 6B; Table S10). On the other hand, BrdU labeling (Fig. 6C; Table S11) indicates that cartilage (within GAG-positive as well as type II collagen-positive regions) in 4-month-old zebrafish contains far greater percentages of proliferating cells than cartilage in 12-month-old zebrafish, and intensive exercise training does not alter the proliferative potential of chondrocytes (Fig. 6D; Table S11). Interestingly, the average percentage of BrdU-positive cells in 12-month-old zebrafish was merely 0.117% (12-month-old), 0.055% (control group) and 0.045% (exercise group) in contrast to the 3.24% in 4-month-old zebrafish, while there were no BrdU-positive cells in many of the 12-month-old cartilage. These results indicate that the cellular renewal is gradually lost as zebrafish aged.

Figure 6 Chondrocytes dynamically turned over in the spinal cartilage.

(A) A representative image of the differential interference contrast (DIC) and TUNEL fluorescent micrograph showed that the apoptotic cell nuclei (pink/white) could be distinguished from normal chondrocyte nuclei (blue as stained by DAPI). The scale bar represents 75 μm. (B) Quantitative analysis showed that the percentage of TUNEL positive nuclei in 12-month-old zebrafish (n = 8) was significantly higher than in 4-month-old zebrafish (n = 7; Mann–Whitney test). However, the apoptotic cell rates were comparable in zebrafish with (n = 8) or without (n = 7) the 2-week exercise training. Data are presented as mean ± SEM. (C) A representative image of the differential interference contrast (DIC) and immunohistochemistry against BrdU fluorescence. The proliferative cell nuclei (pink/white) could be distinguished from static chondrocyte nuclei (blue as stained by DAPI). (D) The percentage of BrdU positive nuclei in 4-month-old zebrafish (n = 8) was significantly higher than in 12-month-old zebrafish (n = 11; Mann–Whitney test). However, the apoptotic cell rates were comparable in zebrafish with (n = 8) or without (n = 8) the 2-week exercise training. Note that the axis scales are different in two different comparisons. Data are presented as mean ± SEM. (E) After pulse-labeling BrdU for 15 days, the 4-month-old zebrafish was cleared from BrdU (chase) for 0, 15 and 30 days to locate the labeling retention cells. The presence of BrdU labeling retention cells tended to decrease with chase time, but the data were not significant (Kruskal–Wallis test). Chase periods were 0 (n = 8), 15 (n = 6) and 30 (n = 6) days. n.s., not significant (P > 0.05); **P < 0.01; ***P < 0.001.

Although the BrdU-positive cells were sporadically distributed in the cartilage with no specific localizations, some of the BrdU-positive cells were located in the peripheral regions where the GAGs and type II collagen were not accumulated (Fig. 6C; Table S11). These cells resided at a location resembling perichondrium with elongated nuclei similar to superficial cells in mammalian articular cartilage. Recent studies imply that these perichondrial superficial cells could serve as stem cells or progenitor cells to provide new cells for chondrocyte turnover (Candela et al., 2014; Karlsson et al., 2009; Li et al., 2017). To assess whether stem cell-like quiescent cells are present in zebrafish cartilage, BrdU pulse-chase was performed in attempting to look for cells retaining the label (Fig. 6E; Table S12). However, we found no BrdU-positive cells in the entire fourth Weberian vertebra in any 12-month-old zebrafish samples (data not shown). In the 4-month-old zebrafish cartilage, the BrdU-positive percentages decreased from 3.24% to 0.58% after a 15-day chase and 1.69% after a 30-day chase (Fig. 6E; Table S12) with no statistical significance. Furthermore, as HMGB2 was previously reported a potential molecular marker for mesenchymal stem cell-like chondrocytes in mouse articular cartilage (Taniguchi et al., 2009), immunohistochemistry using anti-HMGB2 antibody also found no cells being labeled in the entire fourth Weberian vertebra in both 4- and 12-month-old zebrafish (data not shown). Taken together, age indeed affects chondrocyte dynamics and we found no evidence to suggest the existence of cartilage stem cells in mature zebrafish at 12 months of age.

Discussion

In this study, 4- and 12-month-old zebrafish was used to study the effect of age on cartilage homeostasis, especially chondrocyte dynamics. Every zebrafish used in this study demonstrated courtship behavior with female zebrafish and the embryos laid by the female were fertilized indicating the zebrafish, even the 4-month-old ones, were sexually mature. In humans and other mammals, “body maturity” usually comes after “sexual maturity.” Although different body parts vary dramatically, it is generally accepted that the human body reaches full maturity between 20 and 30 years of age and then remains static between 20 and 50 years of age. In our results, it was apparent that the zebrafish continued to grow even after sexual maturity (Figs. 1C and 1E), and therefore body maturity comes after sexual maturity in zebrafish. In line with our result, a previous study indicates that, after sexual maturity, zebrafish continue to grow at least up to 9 months of age (Parichy et al., 2009). The vertebral BMD of zebrafish also showed a significant increase in the control group during the 2-week exercise study period (Fig. 2C). A previous study shows that, although morphologically changed, the BMDs of the fiveth vertebrae of zebrafish are developed at a comparable level at 12, 24 and 36 months of age (Hayes et al., 2013). Taken together, it is likely that the zebrafish reaches full body maturity between 9 and 24 months of age, and probably begins to show signs of aging after 24 months of age without significantly losing BMD.

To evaluate the effect of mechanical loading on cartilage homeostasis in mature zebrafish, a simple intensive-exercise-training system was designed and assembled (Fig. 1A). The maximal swimming speed of 22.4 cm/s is very similar to our test result and indicates that our system could provide intensive exercise training to zebrafish (Gilbert, Zerulla & Tierney, 2014). Our results showed different changes after a 14-day period of intensive exercise training compared to the control group (Figs. 1D and 1F). Among these changes, a significantly shorter body length after a 2-week training program (Fig. 1D) is most surprising and intriguing to us. We attempted to determine the curvatures of the spines using the microCT dataset with no apparent correlative changes (Table S13). One of the tempting speculations to explain this result is the different growth and tone of the musculatures between two groups, as a previous study shows that exercise ability of zebrafish is still trainable at this age (Gilbert, Zerulla & Tierney, 2014).

It is widely accepted that exercise is beneficial to BMD accumulation and can ameliorate the loss of BMD (Shimegi et al., 1994). Furthermore, exercise training in zebrafish larvae stimulates the progress of early endochondral ossification including the Weberian vertebrae, suggesting that the development of the skeletal system indeed is affected by increased mechanical loading (Fiaz et al., 2012). However, we found that a 2-week intensive exercise training negatively affected the BMD accumulation (Figs. 2C and 2D). Interestingly, previous studies indicate that, although some sports positively affect BMD in specific bones, swimming does not positively affect BMD (Bennell et al., 1997; Ferry et al., 2013; Magkos et al., 2007; Maimoun et al., 2013). It is possible that, although the dynamic homeostasis of the skeletal system is affected by mechanical loading, gravity contributes a critical role in this mechanical loading, while buoyancy provided by water minimizes the effect of gravity and hence the BMD of zebrafish is predominantly affected by age and perhaps energy balance (Siccardi et al., 2010). In our study, every zebrafish was individually housed and fed with excessive amounts of food, and therefore the possibility that nutritional insufficiency due to housing or dietary intake could be minimized. In contrast, previous studies suggest that increased exercise in zebrafish promotes catabolic genes such as citrate synthase or nuclear respiratory factor-1 (Liu & Wang, 2013; McClelland et al., 2006). Therefore, we speculate that our intensive exercise training caused a surge in catabolism and in turn hindered the accumulation of BMD and general body mass. Although the BMD in the fourth vertebrae was indeed affected by the exercise training, none of our results showed any difference of cartilages between exercise and control groups. Considering that anterior one-third of the body stays rigid during an adult zebrafish swimming (Fontaine et al., 2008; Muller, Stamhuis & Videler, 2000), the pre-caudal vertebrae, including the fourth Weberian vertebrae, are probably not bearing the mechanical load in a similar way as a mammalian articular cartilage during exercise. Therefore, despite that the cartilage in this area is affected by exercise in zebrafish larvae (Fiaz et al., 2012), it is possible that this model did not provide sufficient mechanical load to cartilage and zebrafish vertebral cartilage was not affected by swimming in 12-month-old zebrafish.

In our observations, type II collagen was more prominently stained in the cartilage margins (Figs. 4A–4C), while GAGs were more prominently stained in the cartilage core (Figs. 5A–5C). To our knowledge, this inconsistency was not described in other articular cartilage, and the generally accepted notion suggests that type II collagen, proteoglycans and GAGs intermingle to constitute the ECM of articular cartilage (Van Der Rest & Mayne, 1988). Furthermore, current evidence suggests that collagen fibers in human articular cartilage mature during teenage years with extremely limited turnover and increase after the age of 20 (Heinemeier et al., 2016; Libby et al., 1964). Our results indicate that both the occupying area and the signal density for type II collagen were increased from 4 to 12 months of age (Figs. 4D and 4E). This result supported our previous speculation that the body maturity of zebrafish came between 9 and 12 months of age. On the other hand, while the collagens in the articular cartilage are extremely inert, the GAGs are dynamically metabolized (Heinemeier et al., 2016; Libby et al., 1964; Mankin & Lippiello, 1969). Accordingly, our result indicates that the signal density for GAGs was already saturated in 4-month-old zebrafish (Fig. 5E). During the increase in occupying area (Fig. 5D), the total amount of GAGs might increase in a linear fashion. Previous study indicates that chondroitin sulfate, the predominant type of GAG in articular cartilage, increases in a linear fashion as zebrafish age from 1 to 3 years (Hayes et al., 2013).

The loss of chondrocytes has been considered one of the reasons for the age-related degeneration of cartilage (Barbero et al., 2004; Stockwell, 1967). In the vertebral cartilage, 2- and 3-year-old zebrafish contain more total lacuna area than 1-year-old zebrafish (Hayes et al., 2013). Two possible explanations could be deduced: (1) old zebrafish have more hypertrophic chondrocytes or (2) old zebrafish lost more chondrocytes. In this study, we attempted to perform immunohistochemistry against type X collagen, a marker for hypertrophic chondrocytes (Inada et al., 1999; Mitchell et al., 2013; Vijayakumar et al., 2013), but failed to obtain any positive signal. On the other hand, although the total cell count increased significantly from 4 to 12 months of age (Fig. 5G), the percentage of apoptotic cells also largely increased (Fig. 6B) supporting the notion that zebrafish lost more chondrocytes with aging. Furthermore, the 2-week BrdU labeling (Fig. 6D) suggests that active chondrocyte proliferation is correlated with the growth and homeostasis of hyaline cartilage.

Previous studies suggest that cells at synovium, tendon, fat pad, and groove of Ranvier might be the sites for origin of articular chondrocytes (Candela et al., 2014; Karlsson et al., 2009; Ohlsson et al., 1992). Recent lines of evidence suggest that the superficial cells in articular cartilage serve as stem cells to provide new chondrocytes during the juvenile stages (Dowthwaite et al., 2004; Li et al., 2017; Taniguchi et al., 2009). However, there has not been solid evidence to suggest the existence of chondrocytic stem cells in mature articular cartilage. Although zebrafish vertebral cartilage was juxtaposited by a perichondrial-like structure (brackets in Figs. 3C and 3D) similar to superficial cells in the mammalian articular cartilage, we did not see any evidence to suggest that these cells are stem cells, nor did we find any evidence for other cells to participate in cartilage homeostasis. Interestingly, our BrdU pulse-chase study showed that some labeling was retained in cells from 4-month-old zebrafish (Fig. 6E), but none of these cells were found in the vertebral column of 12-month-old zebrafish (data not shown). The current model suggests that BrdU dilution via cell proliferation can sufficiently explain the loss of BrdU signal in the chase experiment (Ganusov & De Boer, 2013; Tough & Sprent, 1994). Considering that proliferative cells do exist in the vertebral cartilage, although at a very low level (Fig. 6D), these proliferative cells might go through multiple rounds of proliferation once triggered. Accordingly, our attempt for immunohistochemistry using previously reported stem cell marker for mammalian articular cartilage, HMGB2, also failed to find any positively stained cells in the cartilage of 12-month-old zebrafish (data not shown) (Taniguchi et al., 2009). Hence, it is possible that the homeostasis of mature cartilage depends on the proliferation of terminally differentiated cells, but not stem cells.

Conclusions

Taken together, the body maturity of zebrafish come much later than sexual maturity. A simple exercise training system for zebrafish was designed and demonstrated that short-term intensive swim exercise does not affect cartilage homeostasis. However, similar to mammalian articular cartilage, the hyaline cartilage of zebrafish exhibits different chondrocyte dynamics between young and more mature stages. These results imply that aging perturbs chondrocyte homeostasis and in turn lead to cartilage degeneration.

Supplemental Information

Supplemental Information 1 Supplementary tables.

Supplementary tables include raw data and experimental details.

Click here for additional data file.

Supplemental Information 2 The raw data derived from microCT for the bone mineral density of the fourth spine in zebrafish.

This dataset includes comma-separated-values (csv) files documenting the results of microCT scanning of the fourth spine of adult zebrafish before and after a 14-day intensive exercise training.

Click here for additional data file.

Supplemental Information 3 Fig S1. Body lengths of 12-month-old zebrafish before and after a 2-week period.

Body lengths of 12-month-old zebrafish were measured in micrographs with ImageJ before (+0d) and after (+14d) a 14-day intensive exercise training.

Click here for additional data file.

The authors would like to thank Dr. Harry Mersmann for proofreading and revising this manuscript, and Dr. Yun-Jin Jiang as well as Dr. Ching-Ho Wu for their constructive discussions of this work. We would also like to acknowledge the technical supports from Dr. Chih-Hsien Chiu on histological preparation, Mr. Ting-Hao Wang and the Imaging Core Facility of Taipei Medical University on high-throughput imaging and analysis, Dr. Wei-Cheng Chang, Mr. Hong-Wen Huang and National Laboratory Animal Center on microCT imaging and analysis, Technology Commons, College of Life Science, National Taiwan University on the scanning electron microscopy, and Ms. Ting-Yu Tseng on confocal microscopy.

Additional Information and Declarations

Competing Interests

Author Contributions

Animal Ethics

Data Availability

The authors declare that they have no competing interests.

Quan-Liang Jian performed the experiments, prepared figures and/or tables, authored or reviewed drafts of the paper, approved the final draft.

Wei-Chun HuangFu conceived and designed the experiments, analyzed the data, contributed reagents/materials/analysis tools, prepared figures and/or tables, authored or reviewed drafts of the paper, approved the final draft.

Yen-Hua Lee performed the experiments, authored or reviewed drafts of the paper, approved the final draft.

I-Hsuan Liu conceived and designed the experiments, analyzed the data, contributed reagents/materials/analysis tools, prepared figures and/or tables, authored or reviewed drafts of the paper, approved the final draft.

The following information was supplied relating to ethical approvals (i.e., approving body and any reference numbers):

All experimental procedures in this study were reviewed and approved by the Institutional Animal Care and Use Committee (IACUC) of National Taiwan University (NTU105-EL-00037) and were performed in accordance with the approved guidelines.

The following information was supplied regarding data availability:

The raw data are provided as Supplemental Files.

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
