# Peer review of "Age, but not short-term intensive swimming, affects chondrocyte turnover in zebrafish vertebral cartilage"

_PeerJ, doi:10.7717/peerj.5739_

## Round 0.1 · original submission · Major Revisions

· Academic Editor

Major Revisions

There are several concerns expressed by the reviewers in regard to study design and validity of conclusions. These require clarification or additional experimentation. One of the main concerns identified by reviewer 2 is in regard to the source of the different zebrafish. Please describe exactly differences and similarities between "purchased" and "acquired" ZB, and reasons for using 2 different sources. Furthermore, both reviewers 1 and 2 have concerns regarding insufficient experimental details.

It is also suggested to conduct a more detailed English grammar review.

·

Basic reporting

While the manuscript is well written overall, the text needs to undergo some language editing.
One consistent error involves the tenses used: the results are reported in the past tense, but often so are the conclusions drawn; in my opinion the conclusions should be in the present tense. For example, in the 6th sentence in the abstract, the text should be 'Our results indicate…' and not 'Our results indicated…'.
There are also other errors of syntax, for instance, in the first line of the paper (Introduction section) "…the most common arthritis…"- a different term should be used, perhaps something like "…the most common pathologic condition of articular cartilage …", or a similar term.

The references are adequate, and so is the structure.

The introduction is clear

The relevance of the results and the conclusions drawn from these results are flawed in parts of the manuscript (see below)

Experimental design

As far as I can tell, the research is within the aims and scope of this journal

The research question is somewhat vaguely defined.

The investigation is adequate in terms of most of the techniques used

Some of the methods (like microCT, BMD determination) are not described in sufficient detail, or are not sufficiently clear

Validity of the findings

The authors are trying to establish a model for mammalian articular cartilage by studying the cartilage present in zebrafish cranial vertebra, but have not established the validity of this analogy: cartilage is present only in small areas in most ZF vertebrae, and the cartilage in the Weberian vertebrae seems very different from mammalian articular cartilage in terms of function and structure.

Intense swimming applies heavy loads mostly to the caudal vertebrae of fish, and therefore the pre-caudal vertebrae (particularly the Weberian vertebrae) are probably not the appropriate model for evaluating the effect of mechanical loading.

It is not sufficiently clear how the authors determined the BMD of the vertebrae of the ZF: the resolution of the scan (9 microns) does not seem adequate for this objective, and relevant scan parameter are not provided (rotation step, exposure time, etc.). Furthermore, the results are very low values compared to values reported in the literature (and in my personal experience).

I find the reported result, of significantly increased BMD and body length after 2 weeks of rest (the control group) extremely surprising, and the reliability of these findings must be better supported. In my experience, BMD values do not change significantly within a 2-week period.

The change in length after exercise is not explained well: does this have to do with the effect on the cartilage? Or more likely: an error of measurement or creation of spinal curvature?

Based on Figure 1 F, G, U and J, the control and exercise groups were not similar at the start of the swimming experiment, in terms of weight and body length, so comparisons may be difficult to make.

Line 235-237: I am not clear as to how the authors reached these conclusions.

·

Basic reporting

Generally the manuscript is well written. There are, however, a few typos in the manuscript which need rectifying; these include one on line 375 (currently reads cartialge).

Given that exercise will impact muscle and as muscle is such a substantial component of the zebrafish trunk (and likely the largest determinant of mass) the manuscript could do with some comment on the likely impact of exercise on muscle and some background literature on this.

Experimental design

Could the authors clarify what they mean by fish for 4 and 12 month experiments being being purchased while the others were acquired? As the fish came from different sources are their genetic backgrounds known and comparable between the groups? Fish of different genotypes and/or those raised differently are likely to have more variation between them than the effects from the exercise itself.

When discussing the effects of exercise are comparisons being made to fish from the same source, if not why not?

It was unclear to me whether weights of the same fish were taken before and after training if so then the specific increase and decreases of weight per fish would be better to present - if not why not? How is it known that the experimental group of would have had the same distribution (or length/weights) as the control group .

More detail should be given to the experiment set up for exercise. It states that maximal resisting swim speeds were identified, but it is unclear what speed the fish were subjected to for the 8 hour regime. Surely not 100% of maximal speed as this is the speed at which they hit the net, but if not then what percentage? Or what speed. While calculating the max speed for each fish is admirable if each fish was swum at a different speed it makes direct comparison between individuals difficult and more detail on the experimental design is needed to assess the validity of the conclusions.

What was the stock density at which the fish were raised? Stock density as well as age can have a significant impact on body length and and even bigger one on weight (fish stocked at low density are generally fatter).

The body length decreases seem surprising in a short period - can the authors comment of what they think is occurring - loss of intervertebral distance or contraction of the muscles? Are these comparisons between the same fish (i.e. each fish measured before and after exercise)? What do the authors think would happen if the fish were then allowed to continue without exercise - would the fish regain their length or remain shorter?

Validity of the findings

Most of my points relate to the experimental design above, without more detail on the exercise regime and the source of the fish it is hard to judge the validity of many of the findings as it is somewhat unclear what is being compared.

For example how is it the case that in Fig. 1H exercised fish are heavier than controls, yet, in Fig 1J they have lost mass? This doesn't appear to make sense.

For example are the 12 month control and exercised fish the same ones measured and weighed at 4 months, if not why not?

---

## Round 0.2 · Major Revisions

· Academic Editor

Major Revisions

Although many of the previous concerns have been addressed, as pointed out by reviewer 1, the evidence for this degree of change in BMD after 2 weeks remains insufficient. Reviewer 1 has suggested the nature of evidence and documentation that is required to substantiate this surprising finding, and the authors are encouraged to provide significantly greater detail.

·

Basic reporting

The article meets the listed standards

Experimental design

The paper describes original research, the research question is well defined and relevant.

I still have concerns about a few of the measurements reported in this manuscript, particularly the validity of process of BMD determination (the explanations provided in the rebuttal are in my opinion incomplete), and the values reported are very low for mature ZF.

BMD determination with a lab-source microCT are tricky, and the authors must provide more information: the images must contain scale bars (missing in Figure 2), explain how many slices were included in each VOI, how they confirmed that the equivalent VOI was examined for each fish, how was the partial volume effect avoided, was the scan done 180 or 360 degrees, etc. I would prefer to see in Figure 2 raw data rather than a heat map with no scale.

The other issue is the measurement of total length, and the surprising shortening of trained fish. Can the authors supply the images measured, and indicate the measurement on these images?

Validity of the findings

I am not convinced that a statistically- significant increase in BMD after a 2-week training period is valid, or that such a training period is likely to result in decrease in body length. Naturally my doubts are irrelevant if these findings can be rigorously supported, but I do not feel this was accomplished here.

Comments for the author

I recognize the efforts and work done by the authors, and I think that the questions this study evaluates are relevant and interesting. I think that the paper is generally well written, and much data is provided. Certainly data on the impact of aging on the skeleton are adequately documented.

However I cannot support publication of these unexpected results of the effect of a mere 2-week training on BMD and total body length without more convincing validation.

---

## Round 0.3 · Minor Revisions

· Academic Editor

Minor Revisions

The latest changes you have made to the manuscript and the additional figures have improved the manuscript. However, one reviewer has these remaining concerns:

1. Describe in more detail calibrations for BMD measurements. A log file of the scans (Table S1) is not sufficient for this purpose.

2. Provide a color scale for Figures 2A and 2B, and additional detail on how the measurements were obtained.

·

Basic reporting

OK

Experimental design

Still the BMD measurements are not convincing, calibration is not described, Figure 2 A, 2B do not show the color scale.

Length measurements are also not rigorous.

Validity of the findings

I still feel I am not entirely convinced the findings of BMD and lenth changes after swimming are entirely valid

---

## Round 0.4 · accepted · Accept

· Academic Editor

Accept

Thank you for your attention to the remaining concerns. It is very helpful to have the additional information regarding your methods.

#